# Relationship between Asymmetry Indices, Anthropometric Parameters, and Physical Fitness in Obese and Non-Obese High School Students

**DOI:** 10.3390/ijerph191710533

**Published:** 2022-08-24

**Authors:** Monoem Haddad, Zied Abbes, Rony Ibrahim, Zlatan Aganovic, Khalid Bibi, Johnny Padulo

**Affiliations:** 1Physical Education Department, College of Education, Qatar University, Doha 2713, Qatar; 2Department of Biomedical Sciences for Health, Università degli Studi di Milano, 20133 Milan, Italy

**Keywords:** youngers, obesity, asymmetry, balance, countermovement jump

## Abstract

Impaired balance is associated with an increased risk of lower extremity injuries. The purpose of this study was to investigate the relationship between age, anthropometric measurement, and asymmetry index (AI) in an adolescent high school population. Twenty-six male students (15 ± 1.0 years) were randomly selected. Body composition, measurements of vertical jump height using a countermovement jump test (CMJ), and dynamic single stance balance using the Y-balance test (YBT), were collected over 4 weeks. Hierarchical multiple linear regression analyses were used as dimension reduction techniques in four different blocks to determine valid predictors for AIs. In the first regression analysis, controlling for age, body mass, height, and body mass index (BMI), the regression coefficient (B = 0.383, 95% confidence interval [CI] [0.088, 0.679], *p* < 0.05) associated with body fat indicated that with each additional unit of body fat, the YBT AI increased by 0.383 units. In the second regression analysis, controlling for age, body mass, and BMI, the regression coefficients associated with height (B = −1.692, 95% CI [−3.115, −0.269], *p* < 0.05] and body fat percentage (B = 0.529, 95% CI [0.075, 0.983], *p* < 0.05) indicated that with each additional unit of height or body fat percentage, the CMJ AI decreased by 1.692 units and increased by 0.529 units. Grouping participants based on body fat percentage had a significant effect on the AIs (*p* < 0.05) of the CMJ and YBT. The AI of the CMJ was 15% higher, and that of the YBT was 7% higher in non-obese students than obese students. These findings contribute to the knowledge of the local community and the emerging body of literature on motor skills and competence related to weight in this population.

## 1. Introduction

Adolescent overweight and obesity are related to major health problems, medical conditions, increased risk of adult obesity, and a higher mortality rate [1]. Globally, obesity is considered the fifth leading risk factor for mortality, resulting in about 2.8 million deaths annually [2]. While body mass index (BMI) is ranked the third highest risk factor in the United States of America, it is the greatest risk factor for mortality in Qatar [3]. In a recent cross-sectional study conducted in Qatar as part of the Growth Monitoring Program, the prevalence and characteristics of overweight and obesity among 168,011 high school students (aged between 15 and 19 years) were described [3]. Overall, 43% of the students were overweight or obese (44.4 and 45.6% for boys and girls, respectively). The result of this survey [3] provides stark evidence of a higher-than-expected prevalence of overweight and obese students in Qatar, which was higher than the global prevalence of overweight and obesity (39%), as reported by the World Health Organization [2].

Obese people have lower muscle strength than healthy individuals [4]. Childhood obesity is highly likely to cause deficient gait patterns and lower extremity malalignment, which may lead to an increased rate of overuse injuries [5]. Similarly, increased body mass index (BMI) is related to decreased balance and range of motion, and impaired balance is one of the several risk factors associated with an increased risk of lower extremity injuries [6].

Exercise programs that focus on balance training are associated with reduced injuries [7] and increased neuromuscular power and control during vertical jumps and single-legged drop landings [8]. The Y-balance test (YBT) is one of the most time-efficient tests to evaluate the dynamic limits of stability and asymmetry [9]. Individuals with anterior left/right asymmetries greater than 4 cm on the YBT were 2.5 times more likely to sustain a lower extremity injury [10]. Jumping abilities are recognized as some of the key fundamental movement skills required for students to lead physically active lives [11]. Motor deficits of overweight students are most pronounced in skills that require lifting [12] or propelling the body against gravity [13]. The vertical jump is a common measure of gross motor skill with a documented performance deficiency among overweight students [14]. Although jumping is a common childhood activity often used to assess motor development [15], the jumping mechanics exhibited by overweight students are not well studied. Moreover, the data on the effects of childhood obesity on leg symmetry related to neuromuscular control, flexibility, and strength are very limited.

Furthermore, we hypothesized that body fat percentage (%BF) is expected to negatively affect the AI of physical fitness test measurements, even after considering other anthropometric measures (BMI, height, and body mass), and may be a better predictor of asymmetry in single stance dynamic balance and unilateral vertical jump height than BMI. Therefore, the purpose of this study was to investigate the relationship between age, anthropometric measurements (body mass, height, %BF, and BMI), and asymmetry index (AI) of field physical fitness tests (countermovement jump (CMJ) and YBT) in a student population.

## 2. Materials and Methods

### 2.1. Participants

Twenty-six male students from a local school were randomly selected to be part of this study (Table 1). Participants practiced physical activity as a weekly subject in the school curriculum. Students were free from injury at the time of the experiment and were free from a past injury that would have affected their movement pattern or prohibited them from performing the tasks to their maximum capability. They were also required to abstain from stimulants, depressants, or any other substances, including caffeine, for at least 6 h before measurement and refrain from performing sports or exercise training for 1 day before measurement. All the participants and their legal guardians were informed about the potential risks and benefits associated with the study, and they signed a written informed consent form, agreeing with the protocol procedures and publication of the data. The study was conducted according to the Declaration of Helsinki, and the protocol was approved by the Qatar University Institutional Review Board (QU-IRB 1482-EA/21). All students were fully accustomed to the tests used in this research and were free to withdraw from the study at any time.

### 2.2. Procedures

Data were collected over 4 weeks during the second term (spring 2021). In the first week, all the students attended two orientation sessions on the same day. The first session was dedicated to anthropometric measurements, while the second one involved familiarization with the experimental protocol. The third session was for the actual measurements, in which three tests were performed. A bioimpedance device (ACCUNIQ BC380; SELVAS Healthcare Inc., Daejeon, Korea) was used to measure body composition [16]. The feet and hands of the participants and the eight touch electrodes of the device were wiped with a special wet tissue to improve conductivity. Body mass and height were measured using the incorporated scale and ultrasonic height meter of the device. BMI was calculated by dividing body mass (kg) by height (m^2^). When necessary, dry towels were folded and placed between the thighs and under the armpits to improve inter-segmental resistivity and decrease the margin of error in the estimation of body composition [17]. Fat mass was estimated by the bioimpedance device and was normalized for body mass, and the values were expressed as a percentage of total body weight. Specific age-normalized cut-off values (31% for ≤17 years) were used to assign participants to obese (OB) and non-obese (NOB) groups [18]. Thereafter, photoelectric arrays (Optojump, Microgate, Italy) were used to measure vertical jump height during the CMJ. Three measurements were taken for the following jumping conditions: bilateral and unilateral (for each leg separately) CMJs [19]. Students were instructed to keep their active leg(s) straight in the flight phase of the jump to minimize the margin of error in the embedded algorithm for computing jump height based on the equation of motion. Before the subsequent trial, an inter-trial recovery time of 90 s was given to facilitate near-to-complete muscular recovery. Finally, the YBT was performed using the Functional Movement System Y-balance test kit to measure dynamic single stance balance [20]. Each participant stood on the center footplate with the distal aspect of the right foot at the starting line. While maintaining single leg stance on the right leg, the participant reached with the free limb in the anterior, posteromedial, and posterolateral directions in relation to the stance foot by pushing the indicator box as far as possible. Each participant completed three consecutive trials for each reach direction and alternated limbs between each trial to allow for neuromuscular recovery between trials. The attempts wherein the participant failed to maintain a unilateral stance on the platform, failed to maintain reach foot contact with the reach indicator on the target area while the reach indicator was in motion, used the reach indicator for stance support, or failed to return the reach foot to the starting position under control were disregarded and repeated. The AI was calculated to quantify the inter-limb asymmetry during the unilateral CMJ and YBT using the following formula: (highest performing limb−lowest performing limb)/highest performing limb × 100. The highest performing limb was considered the one to which the largest values are attributed.

### 2.3. Statistical Analysis

Quantile–quantile plots and Shapiro–Wilk tests were used to verify data normality. All the data were normally distributed and presented as mean ± standard deviation (SD). Hierarchical multiple linear regression (HMLR) was first used as a dimension reduction technique in four different blocks to determine valid predictors for AIs of the CMJ and YBT separately. The first block had age as a predictor variable to account for maturation and motor control related variations. In the second block, height and body mass were added to the analysis to account for variations related to growth and anthropometric measurements. %BF and BMI were added to the third and fourth blocks, respectively, to account for obesity. The separating of variables in categories across blocks was performed to reduce the interactions between factors. Assumptions for HMLR were verified before the analysis. Visual inspection of scatterplots showed a linear relationship between the dependent and independent variables. A multicollinearity problem was not identified because the variance inflation factors were less than 3. The Durbin–Watson test, with a value of 2.3, revealed no autocorrelation in the residuals. Standardized residuals were normally distributed, and the Breusch–Pagan test revealed homoscedasticity for the AIs and their significant predictors. Independent *t*-tests were used to evaluate the differences in AIs of the CMJ and YBT between the OB and NOB groups. Statistical analyses were performed using MATLAB (R2020b; Mathworks Inc., Natick, MA, USA) and IBM SPSS Statistics for Windows, version 28 (IBM Corp., Armonk, NY, USA).

## 3. Results

HMLR analyses (Table 2) were conducted to evaluate the prediction of the YBT and CMJ AIs from age, body mass, height, %BF, and BMI. The results of the first block in both regression analyses revealed a model that was not statistically significant (*p* > 0.05). The R^2^ values of 0.009 and 0.018 associated with this regression model suggested that age accounted for 0.9% and 1.8% of the variation in the AIs of the YBT and CMJ, respectively. The results of the second block revealed a model that was not statistically significant for YBT AI (*p* > 0.05) and statistically significant for the CMJ AI (*p* < 0.01). The change in R^2^ values of 0.074 and 0.644 associated with the regression model suggested that body mass and height accounted for 7.4% and 64.4% of the variation in the AIs of the YBT and CMJ, respectively. The results of the third block in both regression analyses revealed a statistically significant model (*p* < 0.05). The change in R^2^ values of 0.242 and 0.089 associated with this regression model suggested that the addition of %BF to the first and second blocks accounted for 24.2% and 8.9% of the variation in the AIs of the YBT and CMJ, respectively. The results of the fourth block in both regression analyses revealed a model that was not statistically significant (*p* >0.05). Additionally, the R^2^ change value of 0.004 and 0.005 associated with this regression model suggested that the addition of %BF to the first and second blocks accounted for 0.4% and 0.5% of the variation in the AIs of the YBT and CMJ, respectively. In the first regression analysis controlling for age, body mass, height, and BMI, the regression coefficient (B = 0.383, 95% confidence interval [CI] [0.088, 0.0679], *p* < 0.05) associated with body fat suggested that with each additional unit of body fat, the YBT AI increased by 0.383 units. In the second regression analysis controlling for age, body mass, and BMI, the regression coefficients associated with height (B = −1.692, 95% CI [−3.115, −0.269], *p* < 0.05) and %BF (B = 0.529, 95% CI [0.075, 0.983], *p* < 0.05) suggested that with each additional unit of height or %BF, the CMJ AI decreased by 1.692 units and increased by 0.529 units. Thereafter, grouping participants based on %BF had a significant effect on the AIs of the CMJ and YBT (*p* < 0.05). The AI of the CMJ was 15% higher, and that of the YBT was 7% higher in the OB group than in the NOB group.

## 4. Discussion

To the best of our knowledge, this is the first study to investigate the use of body fat as a better predictor of AIs of dynamic balance and unilateral vertical jump height than BMI among young students. In accordance with our hypothesis, we found that %BF significantly affected variation in the AIs of the YBT and CMJ by 24.2% and 8.9%, respectively. These results mirror previous studies that demonstrated the negative effect of %BF on the CMJ [21] and YBT [22].

The participants were assigned to OB and NOB groups, and their subsequent grouping based on %BF had a significant effect on the AIs of the CMJ and YBT (*p* < 0.05). However, there was no difference in the BMI values between the groups, which suggests that BMI may be a weak indicator of asymmetry in student populations.

Previously, multiple studies have focused on the negative effect of obesity on the fundamental motor skills of adolescents [13,23]. Specifically, severe deficits in fundamental movement skills, diagnosed as developmental coordination disorder, were significantly linked to a high percentage of body fat [24,25]. Many studies have also shown that obese students exhibit poorer performance in some motor skills [5,6,26]. Furthermore, this inverse relationship could significantly alter motor development [26,27]. These findings concur with our study, showing that obesity, quantified as %BF, is associated with poor gross motor performance and low competence in motor skills and coordination [28,29]. The poorer results of the OB group than those of the NOB group might have been because of the lack of participation in physical activities and low physical capacity [4]. Although obese individuals tend to have greater absolute muscle strength than their healthy counterparts, their relative muscle strength is significantly lower [4].

Additionally, the HMLR analyses conducted to investigate the effect of anthropometric measures on the AIs of the CMJ and YBT among children offered a better insight into the specific role of %BF. Body height and mass were the strongest predictors of CMJ AI scores. The addition of %BF in block 3 significantly affected the variation, suggesting that caution should be exercised when making interpretations related to obesity based solely on BMI among student populations.

Comparison of the AIs of the CMJ and YBT between the two groups revealed that higher body fat was associated with lower scores, indicating that increased %BF would negatively impact jumping and stability. A previous study showed that the values associated with overall motor development, motricity, and balance are significantly lower in obese students than in non-obese students [30]. Obese students demonstrated disadvantages related to the quality of motor skill execution, such as jumping and balancing. Therefore, it is crucial to focus early on improving motor skills in obese students and ensure their participation in sports and other motor activities [31]. Furthermore, local schools must promote physical activities and organize awareness sessions related to daily lifestyle habits, including dietary management, physical activity, and exercise [32].

This study was solely focused on the male sex. Nevertheless, incorporating sex-based analysis in obesity studies may ultimately contribute to improved prevention and treatment, especially considering obesity is prevalent among both boys and girls [33].

This study was conducted on a relatively small number of students, which may limit generalizability. Nevertheless, understanding the causal ordering of the association between students’ motor skills and their weight is insightful [34]. Further studies are warranted to investigate ways to analyze the differences related to gender when examining the relationships between anthropometry and AIs.

## 5. Conclusions

AIs in motor skills involving lower body power development and single leg dynamic balance are affected by the body fat percentage of students rather than their BMI. This result helps physical educators make better decisions that may lead to improved performance and reduced injury rate in sessions where these skills are used.

## Figures and Tables

**Table 1 ijerph-19-10533-t001:** Characteristics of the participants.

Main Outcomes	Mean ± SD (Overall)	Mean ± SD (Non-Obese)	Mean ± SD (Obese)
Participants (n)	26	14	12
Age (years)	15 ± 1.0	15 ± 1.0	15 ± 1.0
Body mass (kg)	78.5 ± 22.9	71 ± 19.7	87 ± 24
Lean mass (kg)	52.6 ± 7.7	50.6 ± 6.8	55 ± 8.1
Fat mass (%)	28.3 ± 12.1	18.5 ± 5.4	39.4 ± 6.2
Height (cm)	170.5 ± 6.9	170.7 ± 7.3	170.3 ± 6.6
* BMI (kg·m^−2^)	26.8 ± 7.6	24 ± 6.8	30 ± 7.3

* BMI, body mass index.

**Table 2 ijerph-19-10533-t002:** Summary of hierarchical regression analysis to determine predictor variables for asymmetry indices of YBT and CMJ.

Model and Predictor Variables	R^2^	$\Delta R^{2}$	ΔF	df
**Model 1: YBT asymmetry index**				
1. Age	0.009	0.009	0.226	(1, 28)
2. Body mass and height	0.083	0.074	0.889	(2, 26)
3. Body fat percentage	0.326 *	0.242	7.539 *	(1, 25)
4. Body mass index	0.330	0.004	0.121	(1, 24)
**Model 2: CMJ asymmetry index**				
**1.** Age	0.018	0.018	0.369	(1, 28)
**2.** Body mass and height	0.662 **	0.644	17.170 **	(2, 26)
**3.** Body fat percentage	0.752 *	0.089	6.121 *	(1, 25)
**4.** Body mass index	0.757	0.005	0.348	(1, 24)

ΔR^2^ = change in R^2^, ΔF = change in F, * *p* < 0.05, ** *p* < 0.01. YBT, Y-balance test; CMJ, countermovement jump.

## Data Availability

The data presented in this study are available on request from JS.

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
