# Peer review of "Relationship between Asymmetry Indices, Anthropometric Parameters, and Physical Fitness in Obese and Non-Obese High School Students"

_ijerph, 2022, doi:10.3390/ijerph191710533_

Round 1
Reviewer 1 Report
Detailed comments
Title: Relationship between age, anthropometric measurements and asymmetry index of fild physical fitness tests in a student population
The work concerns the correlation between anthropometric measurements used to determine the type of body structure and physical fitness tests (CMI, YBT) in a group of students aged 15. In addition, a given physical fitness is tested, so the tests used are research tools, therefore it proposes a change of the title on
Relationship between anthropometric parameters and asymmetry indices and physical fitness on the example of obese and non-obese students aged 15.
Purpose: the title note should be included and the relationship between age and fitness should be deleted as different age groups have not been studied.
Introduction - well defines the research problem related to the difference in physical fitness resulting from the differences in the sizes of anthropometric features.
Material and methods
Participants 2
The statistical characteristics of the surveyed participants included in this part of the work should be transferred to the results, as it was based on the research and is the result of it.
Procedures
Appropriate measurement tools and procedures were used for anthropometric measurements, calculating body composition and measuring vertical stroke (CMJ) and balance test (YBT). The AI asymmetry index was also correctly calculated.
If the correctness of the relationship between age and physical fitness in the text is recognized, the ways of differentiating this feature should be presented in more detail.
The statistical analysis is unqualified, the methods used are appropriate.
Results
Considerations regarding age as a factor in differentiating physical function should be taken into account. especially that there is no description of the results in this regard in the discussion and conclusions
Further studies on a larger population, especially at an earlier age, are advisable to capture the moment when interventions in the field of lifestyle, diet and physical activity are initiated, so that in old age and adults, the effects of poor lifestyle, diet and limited physical activity are limited. Efforts should be made to develop procedures for predicting the effects of anthropometric and fitness features that deviate from the norms that may affect the further quality of life and health.
Author Response
Response to Reviewer 1 Comments
ijerph-1663864
Relationship between age, anthropometric measurements, and asymmetry index of field physical fitness tests in a student population
International Journal of Environmental Research and Public Health
We would like to take this opportunity to thank you very much for all the useful recommendations. We furthermore appreciate your evaluation score. Without exception, all comments and suggestions helped to improve the quality of the manuscript. Each single recommendation has been addressed by the authors. Please find the details below.
Reviewer 1 |
Title: Relationship between age, anthropometric measurements and asymmetry index of fild physical fitness tests in a student population The work concerns the correlation between anthropometric measurements used to determine the type of body structure and physical fitness tests (CMI, YBT) in a group of students aged 15. In addition, a given physical fitness is tested, so the tests used are research tools, therefore it proposes a change of the title on Relationship between anthropometric parameters and asymmetry indices and physical fitness on the example of obese and non-obese students aged 15. Purpose: the title note should be included and the relationship between age and fitness should be deleted as different age groups have not been studied. |
Thank you for highlighting this important point in our study. We considered it and we changed the title |
Material and methods Participants 2 The statistical characteristics of the surveyed participants included in this part of the work should be transferred to the results, as it was based on the research and is the result of it. |
Thank you for the comment. The table was moved to the result section |
Procedures Appropriate measurement tools and procedures were used for anthropometric measurements, calculating body composition and measuring vertical stroke (CMJ) and balance test (YBT). The AI asymmetry index was also correctly calculated. If the correctness of the relationship between age and physical fitness in the text is recognized, the ways of differentiating this feature should be presented in more detail. The statistical analysis is unqualified, the methods used are appropriate. |
We agree with the author that the relationship between age and physical fitness in the text was recognized, which have led us to not reassess it in our statistical analysis and focus more on the relationship between age and motor impairment. The decision to include age in a separate block in the hierarchical multiple linear regression is to account for and isolate variations due to maturation and motor development. Furthermore, this was expected to strengthen our result on the effect of obesity by not allowing for large interactions between all variables. In order to highlight this goal and makes it clearer to the reader, we have added in line 140 the following: Separating variables in categories across blocks was performed to reduce the interactions between factors. |
Results Considerations regarding age as a factor in differentiating physical function should be taken into account. especially that there is no description of the results in this regard in the discussion and conclusions |
Explained previously |
Further studies on a larger population, especially at an earlier age, are advisable to capture the moment when interventions in the field of lifestyle, diet and physical activity are initiated, so that in old age and adults, the effects of poor lifestyle, diet and limited physical activity are limited. Efforts should be made to develop procedures for predicting the effects of anthropometric and fitness features that deviate from the norms that may affect the further quality of life and health. |
Reviewer 2 Report
Interesting idea of this study, my recommendations are the following:
I recommend that the average age of the subjects be mentioned in the abstract.
Lines 40-43 recommend mentioning the conclusions of the study, bibliographic index 3.
I consider that the number of subjects included in the study is very small, consequently the results cannot be generalized. Also apply motor tests without mentioning whether or not the subjects practice physical activity.
Lines 195-196 state that several studies and mention a single bibliographic index. I recommend clarification.
Lines 222-224 sound like a recommendation, so they should be rewritten.
I recommend that you mention at the end of the Discussions section what the limitations and strengths of this study are.
I recommend rewriting the conclusions section focused on results and not on recommendations.
The idea of the study is interesting, but the study has no intervention, only a finding on an extremely small sample. I recommend extending the idea to a larger sample, possibly by age and gender.
Author Response
Response to Reviewer 2 Comments
ijerph-1663864
Relationship between age, anthropometric measurements, and asymmetry index of field physical fitness tests in a student population
International Journal of Environmental Research and Public Health
We would like to take this opportunity to thank you very much for all the useful recommendations. We furthermore appreciate your evaluation score. Without exception, all comments and suggestions helped to improve the quality of the manuscript. Each single recommendation has been addressed by the authors. Please find the details below.
Reviewer 2 |
Thank you for your constructive review. We have made the necessary changes to the manuscript, and we hope that it will be of better quality now. |
I recommend that the average age of the subjects be mentioned in the abstract. |
Thank you for your comment. The average age is added to the abstract |
Lines 40-43 recommend mentioning the conclusions of the study, bibliographic index 3. |
The conclusion of the study is added and highlighted in red |
I consider that the number of subjects included in the study is very small, consequently the results cannot be generalized. Also apply motor tests without mentioning whether or not the subjects practice physical activity. |
Thank you for highlighting this point. The text is amended as follow: Twenty-six male students from a local school were randomly selected to be part of this study (Table 1). Participants practiced physical activity as a weekly subject in the school curriculum. Students were free from injury at the time of the experiment and did not suffer from a past injury |
Lines 195-196 state that several studies and mention a single bibliographic index. I recommend clarification. |
As suggested by the reviewer, this has been fixed: Many studies have also shown that obese students exhibit poorer performance of many motor skills [6,7,27]. |
Lines 222-224 sound like a recommendation, so they should be rewritten. |
Thank you for your comment. The text is amended as follow (Lane 228-229): Further studies are warranted to investigate ways to analyze the differences related to gender when examining the relationships between anthropometry and AIs.
|
I recommend that you mention at the end of the Discussions section what the limitations and strengths of this study are. |
Thank you for your comment. The text is amended as follow:
This study was conducted on a relatively small number of students, which may limit generalizability. Nevertheless, understanding the causal ordering of the association between students' motor skills and their weight is insightful [35]. Further studies are warranted to investigate ways to analyze the differences related to gender when examining the relationships between anthropometry and AIs.
|
I recommend rewriting the conclusions section focused on results and not on recommendations. |
The conclusion is rewritten as suggested by the reviewer focusing on the results. It reads now: AIs in motor skills involving lower body power development and single leg dynamic balance are affected by the body fat percentage of students and not their BMI. This result helps physical educator takes better decisions that lead to improved performance and reduced injury rate in sessions where these skills are involved. |
The idea of the study is interesting, but the study has no intervention, only a finding on an extremely small sample. I recommend extending the idea to a larger sample, possibly by age and gender |
Thank you for highlighting this important point in our study. We will extend this study to a larger number of participants based on age and gender as you suggested. |
Round 2
Reviewer 2 Report
NO COMMENTS